# High-Performance 3D Vertically Oriented Graphene Photodetector Using a Floating Indium Tin Oxide Channel

**DOI:** 10.3390/s22030959

**Published:** 2022-01-26

**Authors:** Jiawei Yang, Yudong Liu, Haina Ci, Feng Zhang, Jianbo Yin, Baolu Guan, Hailin Peng, Zhongfan Liu

**Affiliations:** 1Key Laboratory of Opto-Electronics Technology, Faculty of Information Technology, College of Electronic Science and Technology, Beijing University of Technology, Ministry of Education, Beijing 100024, China; yangjw@emails.bjut.edu.cn (J.Y.); liuyudong@bjut.edu.cn (Y.L.); zhangfeng1995@emails.bjut.edu.cn (F.Z.); 2Jiangsu Provincial Key Laboratory for Advanced Carbon Materials and Wearable Energy Technologies, Soochow Institute for Energy and Materials InnovationS (SIEMIS), College of Energy, Soochow University, Suzhou 215006, China; cihn-cnc@pku.edu.cn (H.C.); zfliu@pku.edu.cn (Z.L.); 3Beijing Graphene Institute (BGI), Beijing 100095, China; yinjb-cnc@pku.edu.cn (J.Y.); hlpeng@pku.edu.cn (H.P.); 4Center for Nano Chemistry (CNC), Beijing Science and Engineering Center for Nanocarbons, College of Chemistry and Molecular Engineering, Peking University, Beijing 100871, China

**Keywords:** vertically oriented graphene, indium tin oxide, photodetectors

## Abstract

Vertically oriented graphene (VG), owing to its sharp edges, non-stacking morphology, and high surface-to-volume ratio structure, is promising as a consummate material for the application of photoelectric detection. However, owing to high defect and fast photocarrier recombination, VG-absorption-based detectors inherently suffer from poor responsivity, severely limiting their viability for light detection. Herein, we report a high-performance photodetector based on a VG/indium tin oxide (ITO) composite structure, where the VG layer serves as the light absorption layer while ITO works as the carrier conduction channel, thus achieving the broadband and high response nature of a photodetector. Under the illumination of infrared light, photoinduced carriers generated in VG could transfer to the floating ITO layer, which makes them separate and diffuse to electrodes quickly, finally realizing large photocurrent detectivity. This kind of composite structure photodetector possesses a room temperature photoresponsivity as high as ~0.7 A/W at a wavelength of 980 nm, and it still maintains an acceptable performance at temperatures as low as 87 K. In addition, a response time of 5.8 s is observed, ~10 s faster than VG photodetectors. Owing to the unique three-dimensional morphology structure of the as-prepared VG, the photoresponsivity of the VG/ITO composite photodetector also presented selectivity of incidence angles. These findings demonstrate that our novel composite structure VG device is attractive and promising in highly sensitive, fast, and broadband photodetection technology.

## 1. Introduction

In the field of optoelectronics, photodetectors play an important role and are widely used in various fields of the military and national economy. In recent years, the combination of graphene and photodetectors has become a hotspot and frontier of international research. Photodetectors based on graphene show unique advantages [1] in response speed [2,3], high carrier mobility [1], and wide-band absorption [4]. However, the photoresponsivity of graphene photodetectors is only several mA/W levels [4,5]. The main reasons for the low response are as follows: weak light absorption (2.3%) of monolayer graphene and short lifetime (≈0.3 ps) of photogenerated carriers [4,6]. Therefore, there are two ways to improve the performance of graphene photodetectors. (1) Combining graphene with optical structures, such as optical microcavities [7], plasmas [8,9], and optical waveguides [10,11]; the main purpose of this is to increase the absorption of light by graphene. (2) Coupling graphene with other nanomaterials or nanostructures; the aim of this is to improve optical absorption and photogenerated carriers collection, such as quantum dots [12], nanowires [13,14], heterostructures [4,15,16], and so on. Nevertheless, the reported devices are mainly based on a transferred monolayer or multilayer graphene sheets, and these methods have some drawbacks. First, the copper-catalyzed graphene growth process is incompatible with mainstream silicon-based semiconductor production lines due to metal contamination and high temperatures [17]. In addition, due to the introduction of metal particles and breakage during the transfer process, the interface of graphene is not ideal and is easily contaminated [18].

Compared with conventional graphene sheets randomly laid down on substrate [19], VG is a class of networks of graphitic platelets that are typically oriented vertically on substrate [20]. Therefore, VG not only has the basic properties of graphene, but also exhibits some distinctive characteristics caused by directional arrangement, which makes it significantly different from traditional graphene films in many aspects. With its unique orientation, non-stacking morphology, and large specific surface area, VG possesses superior mechanical, chemical, electronic, electrochemical, and photoelectric properties [20], which can be used in a wide range of applications. In the sensing field, VG is a kind of high-performance sensing material as a result of its unique vertical direction and open structure [21,22,23]. For example, Shun Mao et al. fabricated a field-effect transistor protein detector by growing VG directly on the electrode [24]. Akhavan et al. designed an ultra-high-resolution electrochemical biosensor for VG [25]. The excellent performance of the detector at both high and low concentrations was attributed to the edge defects of VG, which could accelerate the electron transfer between electrodes and DNA. Similar to VG-based biosensors, the vertical and open structure of VG provides a larger contact area for the adsorption of gas molecules, and then realizes highly sensitive gas detection at room temperature [26]. However, VG is seldom reported in the field of photoelectric detection. Gang Wang et al. constructed quantum dot-functionalized VG/Ge heterojunction photodetectors [21]. The photodetector composed of the hybrid architecture exhibits excellent responsivity and detectivity at a wavelength of 1550 nm. Despite the excellent device responsivity, light absorption mainly relies on quantum dots instead of graphene, thus limiting the spectral range of photodetection. Another hybrid structure that incorporates VG and two-dimensional graphene-Gr into Ge-based photodetectors delivers an excellent responsivity of 1.7 A W^−1^ and detectivity 3.42 × 10^14^ cm Hz^1/2^ W^−1^ at a wavelength of 1550 nm [27]. However, the main light absorption layer of the photodetector at 1550 nm is Germanium rather than VG. During the process of growing VG, to eliminate the effect of interfacial strain, a polycrystalline carbon buffer layer is usually formed, and then many defects and bending areas also come into being [28]. Long, exposed graphene edges and the defects mentioned above could accelerate the recombination velocity of photogenerated carriers and reduce carrier lifetime, thus impairing the detection efficiency of VG photodetectors. Hence, for high-quality VG photodetectors, defects are mortal. In order to obtain excellent performance photodetectors, higher absorption efficiency and crystal quality of VG have already become hot research areas. Therefore, to efficiently generate photocurrents and avoid simple heating of the graphene layer, it is necessary to rapidly separate and transfer photoinduced electrons and holes on the sub-picosecond level [4].

In this work, we report a high-performance infrared photodetection based on VG glass-contacted ITO and systematically characterize its optoelectronic properties. Our results demonstrate photoresponsivity as high as 0.7 A/W under irradiation of near-infrared light (980 nm) at room temperature, which is comparable with other graphene devices [29], and surpasses that of previous reports for monolayer graphene photodetectors by at least three orders of magnitude. This dramatic improvement was achieved mainly by choosing VG as the light absorption layer and floating ITO as the transparent current diffusion layer. In contrast to the simple graphene photodetector, photogenerated electrons and holes can be separated efficiently by ITO into opposite electrodes in our composite structure. Due to the non-stacking morphology of VG, photon absorption efficiency can be increased comprehensively. Meanwhile, the floating ITO layer can passivate the defect state, increase carrier lifetime, and further improve detection efficiency. The use of the ITO layer also effectively reduces the response time of the device. Compared to normal VG photodetectors, rise time (τ_rise_) decreased from 9.5 s to 5.8 s, and fall time (τ_fall_) decreased from 15 s to 6.0 s. In addition, the features of VG are vertical orientation on the substrate and non-agglomerated morphology with a high specific surface area. Consequently, the VG/ITO photodetector allows laser source irradiation angle selectivity. Our composite architecture structure provides a novel strategy for designing high-performance optoelectronic devices based on VG.

## 2. Materials and Methods

### 2.1. Device Fabrication

Through dc-PECVD, VG was grown on 10 cm × 10 cm soda-lime glass for 8 min at 580 °C. Acetone solution, ethanol solution, and deionized water were used to clean VG samples of 1 × 1 cm^2^ in turn. ITO films were deposited on the of surface VG by magnetron sputtering at 100 °C with an Ar flow rate of 50 sccm. Next, the ITO films’ channels were etched, using photoresist as a mask and wet etching of the ITO films with (dilute hydrochloric acid: water = 1:3) solution. After excess indium tin oxide channel was etched, the size of the ITO films’ channel was 70 × 320 μm^2^. Redundant VG was removed by dry etching with oxygen as etching gas. Subsequently, 15 nm Ti and 120 nm Au electrodes were formed using photolithography and metal wet etching. Finally, the completed devices were annealed at 450 °C for 35 s in a vacuum chamber to make the ITO film transparent, reduce resistance, and eliminate residual water molecules from the perfect interface of VG. For low temperature measurement, a sample was placed in an enclosed chamber filled with liquid N_2_ to create a cold environment.

### 2.2. Characterization and Analysis

A Hitachi S-4800 UHR FE-SEM was used for scanning electron microscope characterization of the device. To obtain the transmission of the VG sample, we measured it with a spectrophotometer (Hitachi u-4100). Electrical characteristics and photoresponsivity were measured using a semiconductor parameter analyzer (Agilent B1500A) at room temperature. For illumination during photoresponse measurements, 980 nm and 1550 nm semiconductor lasers (A-0004, HOYATEK) were coupled into a single mode fiber to form a beam with spot size radius of ≈5 mm as a light source. The power values of the output laser were taken by a manually fixated power meter (PM20C, THORLABS) placed above the devices before each electrical measurement.

## 3. Results

Figure 1a shows the three-dimensional schematic illustration of the VG/ITO composite structure photodetector. The composite structure comprises VG growing on traditional soda-lime glass and ITO covering the surface of the VG. The details of VG/ITO photodetector fabrication are described in detail in the Experimental Section. Figure 1b is the cross-sectional view of the composite structure photodetector. The working principle of the VG/ITO composite structure can be understood through the photodetector schematic in Figure 1b. As the light absorption layer of the photodetector, VG dramatically maximized the detecting surface area, thus increasing the light absorption of the photodetector. It is known that ITO is one of the most widely used transparent conducting oxides. ITO functions not only as the carriers’ transport channel but also with less light absorption. When the light source irradiates the photodetector, the VG channel produces photogenerated carriers, which enter the ITO film through the VG channel. Because the ITO film has excellent electrical conductivity, it can significantly reduce the transmission resistance of photogenerated carriers and can be used as a fast carrier’s transmission channel. With the increase in carriers’ transmission speed, it can reach the corresponding electrodes faster; that is to say, it can enhance the lifetime of photogenerated carriers. Therefore, our photodetector can generate more photocurrent and higher photoresponsivity at a certain bias voltage V_bias_. The number of VG layers used is 9–11 in this paper, and the work function of intrinsic VG is about 4.43 eV. The work function of VG placed in the air is close to that of the ITO (4.55 eV), and it is easy to absorb water and oxygen in the air, making VG exhibit p-type transport characteristics. As a result, VG contact with ITO cannot satisfy the conditions for the formation of the Schottky junction, as shown schematically in Figure 1c. When incident light is irradiated on the surface of the photodetector, the absorption of VG in visible to infrared bands is much larger than that of the ITO layer, so VG absorption dominates in the VG/ITO composite system. Therefore, most of the light could pass through the ITO layer and then reach the VG surface. Valence electrons generate immediately after the VG absorbs photon energy and transfers it to the conduction band, forming photogenerated electron–hole pairs. Under external bias, the photogenerated electron–hole pairs will be separated, and then holes and electrons are more easily transported from the inside of the ITO to the external circuit due to the presence of a large number of surface defects and edge defects in the VG.

First, we carried out the basic characterizations of the VG and VG/ITO structure. VG on glass substrate was grown by plasma-enhanced chemical vapor deposition (PECVD) using methods depicted in previous reports [30]. Figure 2a shows the SEM image of the as-prepared VG, which confirms that highly dense graphene nanosheets were standing vertically on the glass surface. The height of a single VG nanosheet was estimated to be about 80 nm, consisting of 9–11 graphene layers with an interlayer spacing of ~0.34 nm [30]. Figure 2b shows the SEM image of VG/ITO structure; the ITO film fluctuates with the VG nanosheets and completely covers the surface of VG. It can be seen from Figure 2b that the thickness of ITO is about 80 nm. Figure 2c shows that the VG nanosheets were further characterized by using Raman spectroscopy. Two main peaks, D peak (≈1300 cm^−1^) and G peak (≈1600 cm^−1^), can be observed from the VG on the glass. The Raman spectrum indicates the formation of graphene films with a D-to-G peak intensity ratio of 2.65, in which the high D peak can be attributed to the presence of defects, wrinkles, or ripples on the graphene in Figure 2c. The optical microscope image of the VG/ITO structure photodetector is shown in the insert. The red dotted area is the VG/ITO composite structure, and the size of the VG/ITO films channel is 70 × 320 μm^2^. The optical transmittance and absorptivity of the ITO, VG, and ITO/VG films were separately prepared on glass substrates and measured with a UV-NIR absorption spectrophotometer. As illustrated in Figure 2d, in the visible to the near-infrared band, the maximum transmittance of the three thin films is ITO, followed by VG and VG/ITO, showing that ITO is almost transparent to light. The transmittance of VG increases gradually with the increase in wavelength, while the transmittance of VG/ITO films increases with the increase in wavelength in the visible band and remains stable in the infrared band at about 79.4% (980 nm). The absorptivity of the three films is shown in Figure 2e, and the highest absorptivity of the three films is the VG/ITO composite structure. The ITO film was grown by magnetron sputtering, which is dark yellow. To increase transmittance, high temperature annealing is required. After annealing, the ITO film cannot be completely transparent and still has certain absorption in the infrared band. Therefore, one of the reasons why ITO/VG composite structures can improve the absorption of the infrared band is that the ITO film has a small amount of light absorption. Furthermore, the ITO serves as a charge transport layer, which conduces to the separation and transport of photogenerated electron–hole pairs in VG. Hence, the combination of ITO and VG could effectively raise the light absorption of VG, especially in the visible and near-infrared bands.

Then, we turned our attention to the photoresponse of our VG/ITO composite structure device in the near-infrared band. For comparison, time-dependent photocurrents of simple VG and VG/ITO composite structures at different illumination powers were respectively measured at room temperature (300 K) as the laser was periodically turned on and off for V_bias_ = 0.1 V. Near-infrared light sources (λ = 980 nm) were used to excite the samples. From Figure 3a,b, we found that the absolute value of time-dependent photocurrent |I_ph_| (|I_ph_| = |I_light_ − I_dark_|) increases significantly with increasing incident laser power density for the two devices. In particular, the variations of photocurrents in the VG/ITO composite structure are more obvious and much stronger (5.8 times higher) than those of the simple VG at the same irradiation conditions. This is mainly because the ITO covering on the VG could effectively act as a carrier transmission channel, speeding up the separation and diffusion of photogenerated electrons and holes in the VG and then inducing the increase in photocurrent. A typical transient light response cycle is shown in Figure 3c; we have further examined the photo switching behavior of the two devices at a bias of 0.1 V (measured at the highest P_in_ = 126.6 µW). The response time of a photodetector is a very important parameter for the practical application of the device. The photocurrent curve of the devices with time can be divided into response time (rise time, τ_rise_) and recovery time (fall time, τ_fall_). It can be seen from Figure 3c, under the irradiation of pulsed laser, that the rise time of the device is far less than the fall time, so the slow fall time determines the final response speed of the device. The response time is defined as the time for the photocurrent to increase from 10% to 90% of peak current, and the recovery time is identified analogously [31]. For the simple VG photodetector, the τ_rise_ and τ_fall_ of photocurrent is 9.9 s and 15 s, respectively. The reason for the longer fall time of the VG photodetector may be that VG is composed of a horizontal buffer layer in contact with the substrate, and graphene nanosheets grow vertically on the buffer layer. Once the light illuminates the VG surface, the graphene nanosheets absorb light energy to produce photogenerated carriers, which transmit through the buffer layer when the carriers flow to the corresponding electrodes. The buffer layer is mainly composed of amorphous carbon or carbide. Amorphous carbon has more defects and greater resistance, which adds to the probability of carrier scattering, affects carrier transport, and thus increases the response time of the VG photodetector. Compared with the VG photodetector, the τ_rise_ of the VG/ITO composite device is 5.8 s and the τ_fall_ is 6.0 s, respectively, which is obviously superior to the former. Due to the addition of floating ITO, photogenerated carriers could enter the ITO thin films through VG. Since floating ITO films possess excellent electrical properties, they not only make up for the defects of VG by increasing the transmission velocity of the photogenerated carriers, but they also facilitate them reaching the corresponding electrode more quickly and then reduce the response time of the photodetector.

Using the data in Figure 3, we calculated the absolute value of photocurrent (|I_ph_|) and plotted its dependence on incident laser power at room temperature in Figure 4a. For all P_in_, the absolute value of photocurrent increases nonlinearly with the increase in optical power. Under the incident laser power of 126.6 µW, the absolute value of the photocurrent of the VG detector is 2.64 μA, while the VG/ITO device is 15.28 μA, significantly higher than the former. One of the most important features of the photodetector is photoresponsivity (R), defined as the ratio of photocurrent to incident laser power, R = I_ph_/P_in_. Figure 4b demonstrates the incident laser power dependence of R acquired at a bias voltage of 0.1 V under different light intensities for the two devices. Obviously, the VG/ITO composite detector presents superior photoresponsivity to the VG one. At a fixed illumination intensity (126.6 μW), the VG/ITO photodetector reaches a photoresponsivity of 120 mA/W, 5.7 times higher than the VG device. The huge improvement of photoresponsivity may result from the floating ITO layer. As carriers transport channel, the ITO on the VG surface extends the lifetime of photogenerated electron–hole pairs in VG, accelerates their separation, restrains the recombination of charge carriers, and then enhances the photoresponsivity of the composite device. To further investigate the basic characteristics of this photodetector, we extracted the bias voltage (V_bias_) dependence of the |I_ph_| (Figure 4c) and calculated the corresponding photoresponsivity of the two devices (Figure 4d). We found that a higher responsivity can be readily achieved by applying a larger bias voltage. Due to a higher bias voltage, it is beneficial to the separation and orientational transportation of photogenerated carriers under a specific excitation power. The VG/ITO composite photodetector shows a remarkable responsivity of about 700 mA/W at 1 V bias voltage, well above that of the VG device (157 mA/W).

Operating temperature has a significant effect on carrier mobility, thermal kinetic energy, and junction characteristics, so the temperature dependence of photodetector performance is a key characteristic in practical application. The photoresponses of two photodetectors are also studied at low temperature. Figure 4e,f show the measured photocurrent and photoresponsivity of the two photodetectors when working at temperatures as low as 87 K. Interestingly, the photoresponse increases notably as the temperature drops for the VG photodetector. For example, under the same conditions (illumination power and bias voltage), the |I_ph_| and R are 21 μA and 126 mA/W respectively, at 300 K. Contrastingly, when the temperature falls to 87 K, the corresponding |I_ph_| and R instead increase to 30 μA and 240 mA/W, respectively. This phenomenon is probably due to the metal–VG contact resistance decreasing monotonically as the temperature declines [32]. However, the VG/ITO structure shows that the photocurrent and photoresponsivity both decrease noticeably as the temperature becomes lower. Once the temperature drops from 300 to 87 K, the corresponding |I_ph_| and R respectively decrease from 90 μA and 700 mA/W to 53 μA and 415 mA/W. The degradation of device performance is probably due to photogenerated carriers freezing at low temperature, reducing their concentration and the photocurrents’ as well. In addition, Er-Jia Guo found through experiments that the resistivity of ITO film would increase with the decrease in temperature [33]. Therefore, for the VG/ITO photodetector, the resistance of ITO film at low temperature is higher than that at room temperature. The low temperature environment caused the channel resistance of the device to increase and the photocurrent to decrease. Although the performance of our composite device partly decreases as temperature drops, it still outperforms the simple VG device, and tolerances of these parameters are acceptable in practical applications.

The response of the photodetector to the incident light angle directly reflects the position information of the light source, which is of great significance to the photoelectric tracking positioning system. Due to the unique three-dimensional structure and high specific surface ratio of VG, we also measured the photocurrent and photoresponsivity at varied incident angles for the VG and VG/ITO photodetectors. A large area of laser spot was utilized to illuminate the device, and the diameter of the laser spot was about 5 mm, which is far larger than the light absorption area of the photodetector, reducing the error caused by the tilted light source. Meanwhile, we ensured that the distance from the light source to the sample was consistent at different angles. For the measurement, we adopted light sources with wavelengths of 1550 nm, which is mainly used in the daily optical communication band, and fixed the incident laser powers of 100 µW. According to Figure 5, it was found that increasing the angle of the incident light source can lead to greater photocurrent and higher photoresponsivity. This is largely attributed to the distinctive three-dimensional structure and high surface-to-volume ratio of VG, which help to increase the absorption of light. In a certain range of incident angles (0–60°), photocurrent and photoresponsivity gradually increase with the incident angle, and both photodetectors exhibit angular selectivity. Notably, the photoresponse of the VG/ITO composite structure is significantly superior to the simple VG one. Consequently, we believe that the VG/ITO device could present good light response properties in the near-infrared band.

## 4. Conclusions

In conclusion, we have demonstrated a high-performing VG photodetector based on a VG/ITO composite structure. Utilizing VG as the light absorbing layer and ITO as the carrier transport layer, we fabricated a highly sensitive photodetector with photoresponsivity up to 0.7 A/W at near-infrared band, improving photoresponsivity by over an order of magnitude compared with simple VG photodetectors at room temperature. In addition, the fast response speed of our composite detector is superior to the VG detector. Meanwhile, the device is also able to work normally in a wide range of ambient temperatures (as low as 87 K) with a lower bias voltage. Additionally, the fabrication process is uncomplicated and repeatable, compatible with current semiconductor processes, and therefore easily prepared on a large scale. Our work proposes a novel method to realize high performance optoelectronic devices based on VG material.

## Figures and Tables

**Figure 1 sensors-22-00959-f001:**
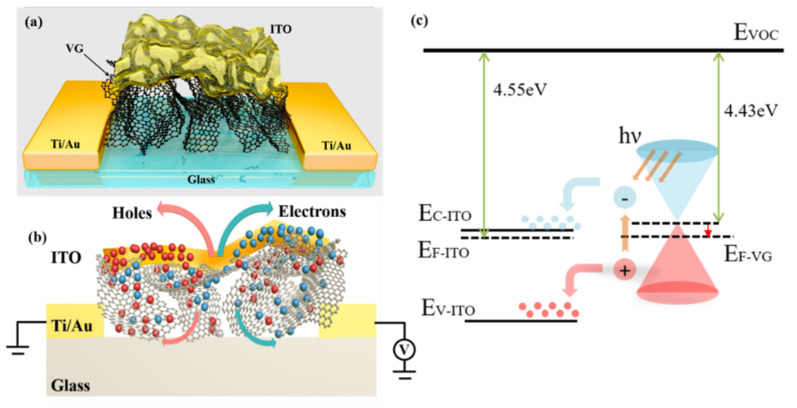
The VG/ITO composite structure photodetector. (**a**) Threedimensional schematic view of the composite photodetector. (**b**) Cross-sectional view of the VG/ITO composite structure photodetector together with electrical connections used to characterize the device. Blue balls represent electrons, and red balls represent holes. (**c**) Schematic of charge transfer at VG/ITO interface.

**Figure 2 sensors-22-00959-f002:**
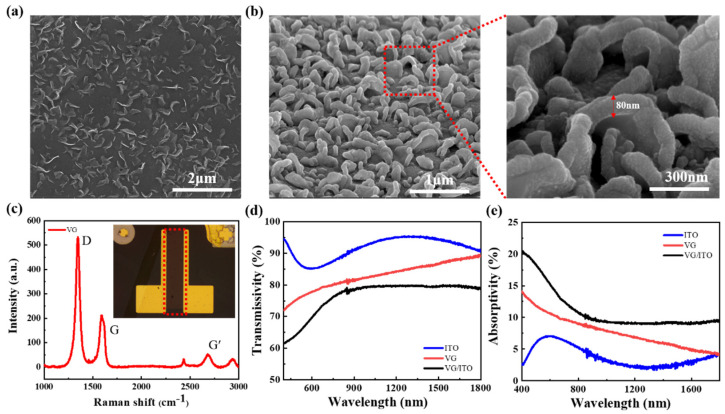
Characterization of VG and VG/ITO. (**a**) SEM images of VG on glass. (**b**) SEM images of VG/ITO on glass. (**c**) Raman spectroscopy of VG. The inset in (**c**) shows the optical image of the VG/ITO composite structure photodetector. (**d**) The transmittance of three films (ITO, VG, and VG/ITO) measured by UV−NIR absorption spectrophotometer. (**e**) The absorptivity of three films (ITO, VG, and VG/ITO).

**Figure 3 sensors-22-00959-f003:**
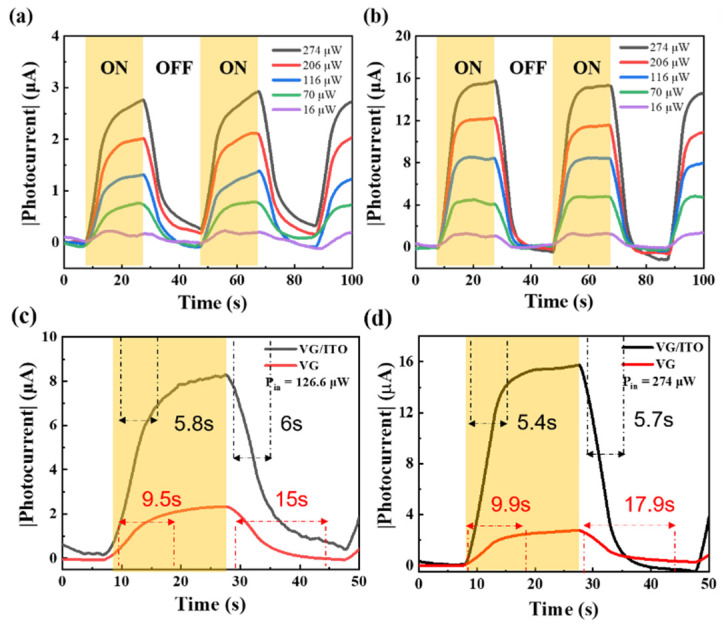
Transient photoresponse of VG and VG/ITO photodetector. (**a**) Time-resolved photoresponse of the VG photodetector. |I_ph_| as a function of time at V_bias_ = 0.1 V and various incident laser powers Pin (from 30.8 to 126.6 µW). (**b**) Time-resolved photoresponse of the VG/ITO photodetector. (**c**,**d**) Enlarged image of one cycle of time-resolved photoresponse of the VG and VG/ITO photodetectors, showing response and recovery times. (**c**) P_in_ = 126.6 µW. (**d**) P_in_ = 274 µW.

**Figure 4 sensors-22-00959-f004:**
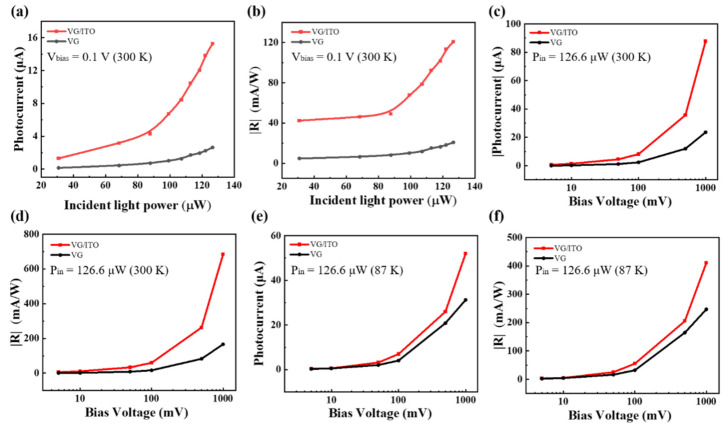
Dependence of photocurrent and photoresponsivity on bias voltage and light power. (**a**) |I_ph_| as a function P_in_ is shown for a fixed bias voltage (V_bias_ = 0.1 V). (**b**) The dependence of photoresponsivity (|R|) on P_in_ for a fixed bias voltage (V_bias_ = 0.1 V) as shown in (**a**). (**c**) |I_ph_| at various bias voltages V_bias_ (from 1 to 1000 mV) for a fixed incident laser powers (P_in_ = 126.6 µW) for the two photodetectors (VG and VG/ITO). (**d**) Photoresponsivity (|R|) as a function of V_bias_ for the three photodetectors. (**e**) |I_ph_| at various bias voltages V_bias_ (from 1 to 1000 mV) for a fixed incident laser powers (P_in_ = 126.6 µW) for the two photodetectors (VG and VG/ITO) at low temperature (87 K). (**f**) Photoresponsivity (|R|) as a function of Vbias for the two photodetectors at low temperature (87 K).

**Figure 5 sensors-22-00959-f005:**
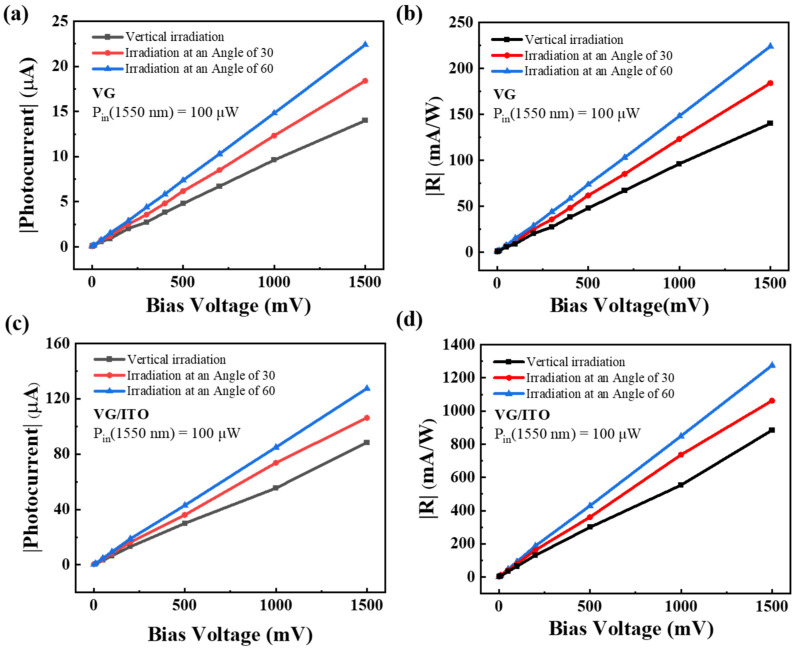
Dependence of photocurrent and photoresponsivity on bias voltage at varied incident angles (0°, 30°, 60°). (**a**) |I_ph_| at various bias voltages V_bias_ for VG photodetector. (**b**) Photoresponsivity (|R|) as a function of V_bias_ for VG photodetector. (**c**) |I_ph_| at various bias voltages V_bias_ for VG/ITO photodetector. (**d**) Photoresponsivity (|R|) as a function of V_bias_ for VG/ITO photodetector.

## Data Availability

The raw data required to reproduce these findings cannot be shared at this time as the data also forms part of an ongoing study.

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
