# Peer review of "High-Performance 3D Vertically Oriented Graphene Photodetector Using a Floating Indium Tin Oxide Channel"

_sensors, 2022, doi:10.3390/s22030959_

Round 1
Reviewer 1 Report
Photoelectronic detection in vertically-oriented graphene next to ITO inclusions, that work as the carrier conduction channel, is examined. The structure is measured and characterized in terms of transmission and absorption while the transient photoresponse is represented for ON/OFF operation at various input laser powers. In addition, the photocurrent and reflectivity are measured for increasing incident light power and bias voltage. The case of oblique incidence is also investigated.
The paper deals with an interesting topic but, due to several weak points, it cannot be published at a journal as selective as Sensors in its present form.
- Where is the sensing and what does the proposed structure sense? Graphs with respect to the frequency or other sensing variables are not provided.
- The authors should define a combined metric taking into account both the reflectivity and photocurrent before and after the ITO installation. In this way can have just a single number to represent and maximized with respect to design parameters.
- The authors should comment on the perspective of employing multiple unit cells like the one examined and on how the collective response will change like in similar configurations [1,2].
- What is the margin for the authors to employ semi-analytical models to solve their device? For example, if the shape of the graphene flakes was canonical like sphere [3,4].
- Can the authors simulate the device numerically and provide spatial distributions of the field intensity throughout their setups?
[1] The role of contact resistance in graphene field-effect devices, Progress in Surface Science, 2017.
[2] Symmetric absorbers realized as gratings of PEC cylinders covered by ordinary dielectrics, IEEE Transactions on Antennas and Propagation, 2014.
[3] Graphene oxide: Hollow spheres, NPG Asia Materials, 2010.
[4] Single-series solution to the radiation of loop antenna in the presence of a conducting sphere, Progress in Electromagnetics Research, 2007.
Reviewer 2 Report
The manuscript entitled “High-performance 3D vertically-oriented graphene photodetector for using a floating indium tin oxide channel (Manuscript ID: 146870)” by Jiawei Yang et al., reports that the broadband and high sensitive photodetector based on vertically oriented graphene/Indium tin oxide composite from the visible to infrared regime (0.7A/W at 980 nm).
The proposed structure of the photodetector uses multilayer graphene grown vertically on the substrate as a light absorption layer, and a well-known transparent electrode, ITO, is applied as a separation and transport layer of a photocarrier. The device is of interest in the field of optical devices in that it improves the performance of the photodetector by separating the carrier absorption layer and the transport layer.
However, the performance of the optical device, such as response time and responsivity, does not show superiority compared to other graphene-based photodetectors, and a clearer explanation of the operating principle seems to be required.
Therefore, I would like to ask the following questions to improve the completeness of the manuscript.
Q1. As described in many previously published graphene and 2D-material based photodetectors, the photoresponse of devices varies very different depending on the power density of incident light and external bias conditions. Moreover, the current responsivity of 0.7A/W on the proposed device does not appear to be superior to that of the devices presented by other groups. Comparison of figure of merits(FOM) for other groups of devices is required.
Q2. In line 192 described on page 5, it seems that a clearer explanation is needed for the content that the ITO/VG composite structure increases the light absorption of VG. (ITO layer is not thought to enhance the inherent light absorption properties of VG.)
Q3. Fig. 3(c) shows the improved photoresponse characteristics of the VG/ITO structure device compared to the VG-based device. However, compared to Figure 3(b), the response time (rising & falling time) of the device is difficult to evaluate. It seems that the reproducibility of the characteristics is not secured.
Q4. In order to analyze the change in the detection characteristics of the device in the low-temperature (89 K) shown in Fig. 4(e.f), a more systematic temperature dependence experiments from low to room temperature is required. (e.g. temperature dependence of the electrical resistance of a device)
Q5. As for the incident light angle dependence in Fig. 5, the detection characteristics (angle selectivity) with the change of the incident angle of the irradiated light on the device are understandable, but as described in the manuscript, the area of ​​the incident channel is smaller than the spot size of the beam. The power density of irradiated beam changes as the angle of incidence increases, making it difficult to accurately measure.
Q6. The wavelength of the light used in the experimental results related to Fig. 3, 4 and Fig. 5 is different (980 nm, 1550 nm). What is the reason?
Reviewer 3 Report
The manuscript is written well and is fit for publication as is.
Author Response
Dear Editors and Reviewers:
Thank you very much for your kindly comments on our manuscript. There is no doubt that these comments are valuable and very helpful for revising and improving our manuscript.
If you have any questions, please be free to contact with us. We are extremely appreciative of your comments.
Yours sincerely,
J. W. Yang & B. L. Guan
Round 2
Reviewer 2 Report
Although the revised manuscript has been reviewed, the reviewer's opinions and answers to questions are not reflected in the revised manuscript (if there are any changes to the content or additional typo corrections, please indicate them), and there seems to be no change.
The responses described in the response letter do not contain additional experimental data or calculation results requested by reviewers. Therefore, as the manuscript did not satisfy rationality in analysis and content to the extent that it was published in this journal, additional revisions are absolutely necessary.
Author Response
Dear Editors and Reviewers:
Thank you for your letter and for the reviewers’ comments concerning our manuscript entitled “High-performance 3D vertically-oriented graphene photodetector using a floating indium tin oxide channel”. These comments are all valuable and very helpful for revising and improving our paper, as well as the important guiding significance to our researches. We have studied comments carefully and have made corrections which we hope meet with approval.
The revised parts are marked in red in the text.
- We changed the grammar. (In line 19 described on page 1)
- We have improved the background and added references. The related works on vertically-oriented graphene photodetectors are listed, and the advantages and disadvantages of these works are described through text. We added reference numbers 28,29, and 36. (In line 75-83 described on page 2)
- We explain the reason that the ITO/VG composite structure increases the light absorption of VG. (In line 200-207 described on page 5)
- We modified Figure 3 (c) and (d), the photoresponse of the VG and VG/ITO photodetector by adopting different optical powers(126.6μW and 274μW), showing response and recovery times.( In line 252 described on page 7)
- We explain why the resistance of VG/ITO composite structure photodetector decreases at low temperature(87K).( In line 296-300 described on page 8)
- We explain the solution to the error caused by the light source tilt. (In line 315-319 described on page 9)
- We explained the reason for irradiating the photodetector with a 1550nm laser. (In line 321-322 and 329-330 described on page 9)
